# Assessing Mechanical Properties of Jute, Kenaf, and Pineapple Leaf Fiber-Reinforced Polypropylene Composites: Experiment and Modelling

**DOI:** 10.3390/polym15040830

**Published:** 2023-02-07

**Authors:** M. M. Alamgir Sayeed, Abu Sadat Muhammad Sayem, Julfikar Haider, Sharmin Akter, Md. Mahmudul Habib, Habibur Rahman, Sweety Shahinur

**Affiliations:** 1Bangladesh Jute Research Institute, Manik Mia Avenue, Dhaka 1207, Bangladesh; 2Manchester Fashion Institute, Manchester Metropolitan University, Manchester M15 6BG, UK; 3Department of Engineering, Manchester Metropolitan University, Manchester M1 5GD, UK; 4Department of Textile Engineering, Jashore University of Science and Technology, Jashore 7408, Bangladesh

**Keywords:** natural fiber, jute, kenaf, pineapple leaf fiber (PALF), polypropylene, composite, mechanical properties, FTIR, TGA

## Abstract

The application of natural fibers is increasing rapidly in the polymer-based composites. This study investigates manufacturing and characterization of polypropylene (PP) based composites reinforced with three different natural fibers: jute, kenaf, and pineapple leaf fiber (PALF). In each case, the fiber weight percentages were varied by 30 wt.%, 35 wt.%, and 40 wt.%. Mechanical properties such as tensile, flexural, and impact strengths were determined by following the relevant standards. Fourier transform infrared (FTIR) spectroscopy was employed to identify the chemical interactions between the fiber and the PP matrix material. Tensile strength and Izod impact strength of the composites significantly increased for all the composites with different fiber contents when compared to the pure PP matrix. The tensile moduli of the composites were compared to the values obtained from two theoretical models based on the modified “rule of mixtures” method. Results from the modelling agreed well with the experimental results. Tensile strength (ranging from 43 to 58 MPa), flexural strength (ranging from 53 to 67 MPa), and impact strength (ranging from 25 to 46 kJ/m^2^) of the composites significantly increased for all the composites with different fiber contents when compared to the pure PP matrix having tensile strength of 36 MPa, flexural strength of 53 Mpa, and impact strength of 22 kJ/m^2^. Furthermore, an improvement in flexural strength but not highly significant was found for majority of the composites. Overall, PALF-PP displayed better mechanical properties among the composites due to the high tensile strength of PALF. In most of the cases, T_98_ (degradation temperature at 98% weight loss) of the composite samples was higher (532–544 °C) than that of 100% PP (500 °C) matrix. Fractured surfaces of the composites were observed in a scanning electron microscope (SEM) and analyses were made in terms of fiber matrix interaction. This comparison will help the researcher to select any of the natural fiber for fiber-based reinforced composites according to the requirement of the final product.

## 1. Introduction

The past few decades have seen the emergence of natural fibers in our daily commodities, in their basic forms or in other geometric textile structures such as yarns, fabrics, and non-woven sheets, as the alternative reinforcements to create composite materials due to the ecological advantages as they offer over the manmade fibers such as glass or carbon fibers [1,2,3,4,5,6,7]. Among the natural fibers, few members of the subgroup of bast fibers such as jute, flax, hemp, kenaf and ramie, and few members of the subgroup of leaf fibers such as sisal and pineapple have attracted particular attention from both the industry and academia. Jute that belongs to the species of Corchorus capsularis (White jute) and Corchorus olitorius (Tossa jute) is known as a lignocellulosic bast fiber due to its high content of lignin (12–13%) together with cellulose (over 61–71.5%) and hemicellulose (13.6–20.4%) [6,7,8,9]. It is the second most important natural fiber after cotton and is mostly produced in Bangladesh, India, and China. On the other way, Kenaf is another lignocellulosic bast fiber that comes from the plants of the species of Hibiscus cannabinus commonly grown in tropical and subtropical Africa and Asia. Its chemical constitution includes 60–80% cellulose, 5–20% lignin, and up to 20% moisture [10]. Nevertheless, pineapple (Ananas comosus) is grown mainly for its fruit, but its non-edible leaves produce a lingo-cellulosic fiber having a composition of 67.12–82% cellulose, 9.45–18.80% hemicellulose, and 4.40–15.40% lignin [4] and it is a waste of pineapple product. Several contemporary research works reported the applications of these fibers as composite reinforcements in both thermoplastic and thermoset plastic matrices for structural as well as value-added product development [3,4,10,11,12,13,14,15,16,17].

A summary of jute, kenaf, and PALF fiber composites is presented in Table 1. Several studies were carried out where the three fibers were incorporated into the PP matrix in different forms (short fiber, long fiber, nonwoven mat) using compression and injection molding and by extrusion in some cases. In general, the fiber content varied from 10 wt.% to 55 wt.%. The mechanical properties of the composites were determined by tensile, flexural, and impact tests to ensure their successful fabrication and ability to perform in applications. When the natural fibers were chemically modified, they promote better interfacial strength or in achieving specific properties such as flammability or hydrophobicity. Observation of the fractured surface by SEM determined interfacial bonding characteristics. In general, the water uptake increases in the composites compared to the pure PP matrix. Not only TGA, DMA but also DSC was tested to observe the thermal performance of the reinforced composites. Appropriate fiber treatment can improve functional characteristics of the resulted composites. After careful reviewing of the current literature, it is clear that varying results on the functional characteristic of the jute, kenaf, and PALF fiber composites are reported due to the variations in fiber content, forms, treatments, and processing techniques. Although most of the investigation studied focused on a single fiber i.e., either jute, or kenaf or PALF, Ng et al. [18] made a comparison on the mechanical properties of the composites based on PP with kenaf and PALF. In general, PALF composites showed better properties than kenaf-based composites and 30 wt.% fiber resulted the best mechanical properties.

However, no studies presented a comparison of the composites with the three selected natural fibers such as jute, kenaf, and PALF-reinforced composites. From the composite material fiber selection point of view of, it is important to prepare different fiber-reinforced composites in identical conditions such as manufacturing process, fiber content, fiber size, forms etc. This would provide a fair comparison to assess the effect of different natural fibers. Herein, this paper aims to compare the structural, mechanical, and thermal properties of jute, kenaf, and pineapple leaf fiber (PALF)-reinforced polypropylene (PP) composites by varying weight percentages of individual fibers using the same fabrication and testing procedure.

The rest of the article is organized in a way that Section 2 provides raw materials and their composite fabrication processes with their related testing methods. Section 3 highlights important outcomes and analyses of the comparative properties of the composites. Finally, Section 4 presents important conclusions drawn from this study.

## 2. Experimental Procedure

### 2.1. Materials

Jute fibers (Tossa jute: Corchorus Olitorius), Kenaf (HC-95 variety: Hibiscus Cannabinus) were collected from Experimental Research Stations of Bangladesh Jute Research Institute (BJRI) and pineapple leaf fiber (PALF: Ananas Comosus) was obtained from Madhupur Thana under Tangail district of Bangladesh. Polypropylene pallets (Brand: SABIC, KSA) were collected from the local market of Dhaka, which were used as a matrix material. The fiber color, density, and mechanical properties of different raw materials are listed in Table 2. From the table, it is clear that the trend of diameter for the fibers is Kenaf > Jute > PALF, density PALF > Kenaf > Jute and tensile strength PALF > Kenaf > Jute.

The single fiber was characterized by tensile testing using an Instron machine integrated with a load of 100 N at a crosshead speed of 2.5  mm/min and the fiber span length of 25  mm following D3822ASTM standard. Diameter of the fibers were measured by a Fineness tester and image analysis.

Figure 1 presents fiber surface morphologies of the three fibers. The surface of raw jute fiber was generally smooth with ridges and irregular cellular nature with some micropores. Whereas the kenaf surface also appeared smoother with ridges. However, the PALF surface was rougher with regular cellular structure. The fiber diameters for jute, kenaf and PALF are approximately 30–50 µm, 50–65 µm, and 20–35 µm respectively.

In this study, polypropylene (PP) was used as matrix material and jute (Tossa variety), kenaf, and pineapple leaf fibers (PALF) were used as the reinforcing materials for fabricating composites.

### 2.2. Composite Fabrication Process

#### 2.2.1. Fabrication of PP Sheets

The PP sheet of 1 mm thickness was fabricated by melting and compressing for pre-weighed PP pellets by the CARVER heat press machine (Carver, INC, Model 4128, Wabash, IN, USA) at 190 °C and a pressure of 1814 kgf for 10 min using a square mold dimension of 300 × 300 × 1 mm^3^. The fabricated PP sheets were then cooled through water flow at room temperature (25 °C) for 15 min.

#### 2.2.2. Fabrication of Composite Laminates

Composites of 4 mm thick were prepared by sandwiching two layers of fibers between three pre-weighted PP sheets as shown in Figure 2. These fibers are carefully placed between PP sheets by ensuring its uniform distribution as far as the mass of the fibers per unit area is concerned. Fiber weights in each composite laminate were measured.

The sandwiched PP sheets were then placed between two steel molds with randomly oriented fibers and heated at 190 °C for 15 min to soften the polymer sheets and impregnated into the fiber at 2268 kgf pressure. Finally, the mold was allowed to cool at room temperature. After that the fiber-reinforced PP composite sheet was removed from the mold plate. The nominal thickness of the composite plate was approximately 3.00 mm.

### 2.3. Fourier Transform Infrared (FTIR) Spectroscopy

Fourier transform infrared (FTIR) spectroscopy was carried out in a Shimadzu 81,001 spectrophotometer at Bangladesh University of Engineering and Technology (BUET), Bangladesh. The transmittance range of the scan was set from 650 to 4000 cm^−1^. To obtain the spectra, the attenuated total reflectance (ATR) mode was employed.

### 2.4. Physical and Mechanical Characterization of Composites

#### 2.4.1. Determination of fiber fractions

The weight of final composite sheet was measured using a weighing balance and accordingly, the fiber weight content (%) in the fiber-reinforced composite was determined using Equation (1).
(1)$$\;Fibre\;weight\;\left(\%\right)=\frac{Total\;fibre\;weight\;in\;the\;composite}{Composite\;sheet\;weight}\times 100$$

Table 3 presents fiber fractions in each composite with the corresponding sample IDs.

#### 2.4.2. Determination of Tensile Properties

Tensile test of fiber-reinforced composite samples was carried out using Instron Universal testing machine (3369 series) equipped with a 5000 N load cell and a cross-head speed of 5 mm/min. Specimens with a nominal dimension of 180 mm × 20 mm × 4 mm for each type of composite were employed during the uniaxial tensile tests used for tensile testing. Standard dumbbell-shaped test specimens were tested according to ASTM D638 standard. Five specimens were tested for each composite to check the test repeatability. The tensile test specimens are shown in Figure 3.

#### 2.4.3. Determination of Flexural Properties

The flexural strengths and moduli of the composite specimens were measured using a three-point bending test according to ASTM D790-02: 2002 test standard in Hounsfield H10 KS, UK. The tests were carried out with a span-to-depth ratio of 16:1 and at a crosshead speed of 2 mm/min. Specimen dimensions for three-point bending tests were 120 × 13 × 4 mm^3^. It should be noted that the edges of composite samples were smoothened by sandpaper in order to avoid stress concentration during the tensile and bending tests. The flexural strength ($\sigma_{f}$) and modulus of the composite samples were determined using the Equations (2) and (3).
(2)$$\sigma_{f}=\frac{3PL}{2bd^{2}}$$
(3)$$E_{f}=\frac{L^{3}m}{4bd^{3}}$$
where *P* is the applied load (N), *L* is the length of support span, *b* is the specimen width (mm), *d* is the specimen thickness (mm), and *m* is the slope of the tangent to the initial straight-line portion of load-deflection curve (N/mm).

#### 2.4.4. Determination of Impact Properties

Impact strength is defined as the ability of a material to absorb energy. The impact strength of composites is directly related to its overall toughness. The composite toughness is affected by interlaminar and interfacial strength parameters. The Izod impact test for un-notched specimens was conducted using an impact-testing machine of HUNG Ta Instrument Co., Ltd. (Taiwan). The load of the pendulum was 4 J. The impact properties were measured according to ASTM D256 test standard. Specimen dimensions for impact strength tests were 90 mm × 13mm× 4 mm. It should be noted that the edges of composite samples were smoothened by sandpaper in order to avoid stress concentration during the Izod Impact tests.

### 2.5. Thermogravimetric Analysis of Composites

The thermal stability of the composite samples was assessed by using the Thermogravimetric Analyzer ELTRA Thermostep (Eltra GmbH, Haan, Germany). TGA measurements were carried out on 40–50 mg sample placed in a platinum pan, heated from 30–600 °C at a heating rate of 10 °C/min in a nitrogen atmosphere with a flow rate of 20 mL/min to avoid unwanted oxidation.

### 2.6. Fractured Surface Analysis Procedure

The fractured surfaces of the composites were observed under a scanning electron microscope (SEM) to analyze the adhesion and interfacial characteristics between the natural fibers and the matrices. The fractured surface was coated with a thin layer of gold to make it conductive. An SEM of model JSM-5600LV from JEOL Ltd. was used at an accelerating voltage equal to 20 kV in secondary electron mode.

## 3. Results and Discussion

The virgin PP and the natural fiber-reinforced composites were characterized for structural, mechanical, and thermal properties. To compare the composites, the theoretical values of Young’s moduli were calculated by two specific theoretical models and compared with the experimental results.

### 3.1. FTIR Analysis

FTIR of three fibers (jute, kenaf, and pineapple), as well as their respective composites, are depicted in Figure 4.

Owing to the cellulose structure of the fibers, broad peaks of OH group were appeared between 3300 and 3400 cm^−1^. The peaks were slightly shifted with the variation of fibers. In case of jute, the OH peak was located approximately 3336 cm^−1^ whereas it was at 3329 cm^−1^ for both the kenaf and pineapple fibers. Similar shifting was also visualized for other important peaks such as C-H, C-C, C-OH, C-H ring, and aromatic C-H plane. Meanwhile, due to a C-H vibration of -(CH3) group, a peak for natural fibers in the region of 2900 to 2920 cm^−1^ was also found with small variations among the fibers for the existence of the methyl group (–CH2) of cellulose and hemicellulose portion [44]. Furthermore, C-O stretching in carbonyl and unconjugated β-ketone appeared at peaks between 1730 and 1740 cm^−1^ due to the presence of xylan in hemicellulose [44]. Moreover, peaks appeared between 1454 and 1375 cm^−1^ indicating the symmetrical bending of -CH3 bending of lignin and other carbohydrate compounds. Another broad peak at around 1000 to 1100 cm^−1^ represents C-C, C-OH, C-H ring, aromatic C-H plane deformation and side group vibration of the fibers because of hemicelluloses and lignin [35]. Similar types of peaks (OH-, CH-C-C) were also reported for untreated and chemically treated jute fibers-reinforced composites [45,46]. On the other hand, PP matrix material, C-H vibration of -(CH3) group appeared in the region of 2900 to 2920 cm^−1^. From the figure, it was clear that FTIR of composite was the superimposed version of the selected fibers and the matrix material. The peaks of the fibers as well as the matrix material both were visible in the composite material. Neither any evidence of additional peak nor significant peak shifting occurred in case of the composites signifying no or lack of strong chemical bond between the fibers and the matrix.

### 3.2. Tensile Strength (TS)

The fracture surfaces of the fiber-reinforced composite are shown in Figure 5.

The tensile test results for the fiber-reinforced PP composites are presented in Figure 6. It was clear from the figure that all the fabricated composites displayed higher tensile strengths than the PP matrix itself. For the jute-PP and kenaf-PP composites, a decreasing trend of tensile strength was noticed with an increase in the fiber content. At 30 wt.% and 35 wt.% fiber content kenaf-PP composite showed better tensile strength than the jute-PP composite. However, at 40 wt.%, the strength of kenaf-PP composite dropped significantly. Although at 30 wt.% and 35wt.% fiber content, PALF-PP composite showed lower strength compared to the other composites, but the PALF-PP composite produced the highest TS at 40 wt.%.

The general trend in tensile strength of the composites for all fiber wt.% can be arranged in the following descending order: jute-PP < kenaf-PP < PALF-PP except the PALF-PP composite at 40 wt.%, which was higher than the 40 wt.% kenaf-PP composite. Similarly, for all composites the tensile modulus decreased with the fiber loading. Among the composites, the jute-PP composite displayed the best stiffness which was more than double the value of the PP.

Benhamadoucheet al. [47] found that recycled jute fabric-reinforced PP composite showed a tensile strength of 30 MPa but a modulus of (4–4.5) GPa, which was two times higher than the tested jute-PP composite in this study. Whereas Shahinur et al. found tensile strength of 30–34 MPa and tensile modulus of (2–4) GPa for jute-PP composite [48]. Akil et al. [49] reported that until 60% kenaf fiber incorporation increased the strength and modulus of the PLA composite, whereas in this study, a fiber loading greater than 30 wt.% the tensile strength and modulus have decreased for the Kenaf/PP-reinforced composite.

In this study, the PALF-PP composites showed lower TS and TM compared to the kenaf-PP composites except the 40% fiber loading condition. However, Feng et al. [50] found higher TS and TM in case of PALF fiber-reinforced composite compared to the kenaf-PP composite. This could be due to the difference in the fiber structure and the adhesion criteria between the fiber and matrix. Furthermore, in another study, Feng et al. [51] also reported that after chemical (NaOH) treatment, strength, modulus, and impact properties were linearly increased for the kenaf-PP and PALF-PP composites where 30 wt.% fiber was incorporated in chopped mode and the method of composite fabrication was the hot press. Berzin et al. [41] found increasing trend of tensile strength and tensile modulus with the PALF incorporation between 10, 20, 30 wt.% in the PP matrix when the composites were fabricated through the twin-screw extrusion method. Gadzama et al. [36] found the pineapple leaf-PP composite tensile properties of PALF-PP composites increased until 30 wt.% after that they decreased. However, in this study, tensile properties were found higher in case of 40 wt.% pineapple leaf fiber-reinforced composites compared to the other composites.

At low fiber content, a small fiber population contributes to low load transfer capacity among the fibers. As a result, accumulation of stress occurs at certain points of the composite. It was evident that higher fiber loading increased the probability of fiber agglomeration within the matrix which produces non-uniform stress transfer and stress concentration promoting crack propagation. Moreover, too many fiber ends encourage micro crack formation in the interface. As a result, the strength and modulus of the composite again decrease. Generally, the toughness of fiber-reinforced polymer composites is dependent on the fiber, the polymer matrix, and the interfacial bond strength [52,53].

### 3.3. Theoretical Calculation of Tensile MODULUS

The tensile modulus of composites is primarily dependent upon the fiber volume fraction [54,55] and it can be predicted based on the modified “rule of mixtures” method. According to Sayeed et al., 2013 [56], the theoretical models of tensile modulus of natural fiber-reinforced composites have been analyzed for predicting the tensile modulus of jute/PP nonwoven composites. Motivated by their work, Miao and Shan [57] and Pan [58] models for short fiber composites were applied for predicting the tensile modulus of jute, kenaf and PALF fibers-reinforced PP composites. Accordingly, Miao and Shan [57] have attempted to predict the tensile modulus of short fiber composites by employing Krenchel fiber orientation factor in the rule of mixtures equation, as shown below.
(4)$$E_{C}=\eta_{0}\eta_{l}V_{f}E_{f}+\left(1-V_{f}\right)E_{m}$$
(5)$$\eta_{0}=\sum a_{n}cos^{4}\theta_{n}$$
where *E_f_, E_m_*, and *E_c_* are the modulus of fiber, matrix, and composite, respectively, *η_l_* is a length efficiency factor, *η_o_* is a Krenchel factor related to fiber orientation, *V_f_* is the fiber volume fraction, and *a_n_* is the fraction of fibers with orientation angle *θ_n_* with respect to the loading direction. Alternatively, Pan has proposed to replace fiber volume fraction (*V_f_*) in rule of mixtures method by fiber area fraction (*A_f_*) that accounted for the fiber orientation and the direction of the cross-section. In order to compute the direction dependence of composite tensile properties, the fiber area fraction and the area fraction of matrix are defined on a plane (Θ,ΦΘ,Φ), as shown below.
(6)$$A_{f}\left(\Theta ,\Phi \right)+A_{m}\left(\Theta ,\Phi \right)=1$$
where $A_{f}\left(\Theta ,\Phi \right)$ and $A_{m}\left(\Theta ,\Phi \right)$ are the fiber and matrix area fractions on a plane defined by polar angle (Θ) and base angle (Φ). In addition, a relationship between the fiber area fraction $A_{f}\left(\Theta ,\Phi \right)$, and the fiber volume fraction has also been formulated [58], as illustrated in the Equation (7).
(7)$$A_{f}\left(\Theta ,\Phi \right)=\Omega \left(\Theta ,\Phi \right)V_{f}$$
where $\Omega \left(\Theta ,\Phi \right)$ is the probability density function of fiber orientation projected on the plane defined by $\left(\Theta ,\Phi \right)$. Accordingly, the rule of mixtures has been modified as shown below [58].
(8)$$E_{C}=\eta_{l}A_{f}\left(\Theta ,\Phi \right)E_{f}+\left(1-A_{f}\left(\Theta ,\Phi \right)\right)E_{m}$$

Alternatively,
(9)$$E_{C}=\eta_{l}V_{f}\Omega \left(\Theta ,\Phi \right)E_{f}+\left(1-V_{f}\Omega \left(\Theta ,\Phi \right)\right)E_{m}$$

In this study, the short fibers reinforced in the composites are hand laid in two layers in between polypropylene films and randomly oriented in the composites in order to obtain the “quasi-isotropic” structure. Therefore, these fiber structures can be easily presumed to be “quasi-isotropic” type of structure having fibers orientated randomly in three dimensions (3D). Accordingly, the Krenchel fiber orientation factor and the probability density function for ideally randomly oriented short fiber structures given in the Eqs. 5 and 7 are 0.2 and 0.159, respectively [57,58]. Moreover, the length efficiency factor of fibers can be ignored, i.e., $\eta_{l}=1$, as the fibers in the short fiber structures have a very high aspect ratio [58]. Hence, Equations (4) and (9) were used for computing the tensile modulus of ideally randomly orientated short fiber composites based on the work of Miao and Shan, 2011 and Pan, 1994 are given below.
(10)$${\left(E_{C}\right)}_{Miao\;and\;Shan\;}=0.2V_{f}E_{f}+\left(1-V_{f}\right)E_{m}$$
(11)$${\left(E_{C}\right)}_{Pan}=0.159V_{f}E_{f}+\left(1-0.159V_{f}\right)E_{m}$$

Accordingly, the tensile modulus of short fiber composites has been predicted based on Equations (6) and (8) and subsequently, compared with the experimental tensile modulus of jute, kenaf, PALF-reinforced PP composites. The modulus of 30/70 jute/polypropylene, 35/65 jute/polypropylene and 40/60 jute/polypropylene composites were observed to be 2.11, 2.03, and 1.87 GPa, respectively. Similarly, the moduli of 30/70 kenaf/PP, 35/65 kenaf/PP, 40/60 kenaf/PP composites were 1.89, 1.71, and 1.46 GPa and for 30/70 PALF/PP, 35/65 PALF/PP, 40/60 PALF/PP composites, the moduli of these composites were 1.64, 1.53, and 1.67 GPa respectively. In general, a good agreement has been obtained between the experimental and the theoretical results of tensile modulus of short fiber composites obtained by the Pan model, as shown in Figure 7. On the other hand, the model defined by Miao and Shanfor predicting the tensile modulus of short fiber composites has clearly overestimated the experimental results. Similarly, Sayeed et al. has predicted the tensile modulus of nonwoven jute/PP composites for different stacking sequences of nonwoven layers [56].

### 3.4. Flexural strength (FS)

Generally, structural materials that have high tensile strength also perform well at the bending load due to good interaction with the fiber and matrix. In this regard, flexural properties were investigated and shown in Figure 8. From the figure, it was clear that at 35 wt.% fiber content, both jute-PP and kenaf-PP composites showed the best flexural strengths in contrary to the highest FS found for the PALF-PP composite at 40 wt.%. An increasing trend of strength from 30 wt.% to 35 wt.% and decreasing trend from 35 wt.% to 40 wt.% was seen for the jute and kenaf composites, whereas an inverse trend was found for the PALF composites. In general, the kenaf-PP composites displayed poorer flexural strengths compared to the others whereas jute-PP composites showed quite consistent behavior for all three fiber contents. However, in terms of flexural modulus, a significant improvement was noticed for all composites compared to the matrix. The general trend in flexural modulus of the composites for all fiber wt.% can be arranged in the following descending order: jute-PP < kenaf-PP< PALF-PP similar to the tensile modulus. However, no clear correlation was found between the flexural strengths and moduli except at 35 wt.%.

Force vs. extension curves for the composites during flexural testing are presented in Figure 9. From the extension plot it was clearly observed the extensions before failure increased with up to 35 wt.% fiber content but at 40 wt.% the extensions were similar to the matrix material.

Although all composites showed higher or similar flexural strength values compared to PP, but the respective increase in FS was not as high as that in the tensile strength. The poor flexural strength results for kenaf-PP composite could be related to relative lower fiber strength and/or inadequate interfacial adhesion.

### 3.5. Impact Strength (IS)

Impact strength of the fabricated jute, kenaf, and PALF-reinforced composites are shown in Figure 10. A general trend of increasing impact strength was noticed in all composites with the increase of the fiber content. Among the composites, the best and worst impact properties were found for PALF-PP and kenaf-PP composites respectively possibly due to their individual fiber strength characteristics as shown in Table 1. The results emphasized the fact that all the composites will absorb higher energies before breaking than that of the PP matrix.

### 3.6. Thermal Characteristics

The effects of temperature on the mass changes of the jute, kenaf, and PALF-reinforced composites are presented in Figure 11.

There are three significant regions of weight loss due to a rise in the operating temperature. According to Shahinur et al. [53], the initial low temperature weight loss of the composites is due to the removal of moisture from the composites, major weight loss due to degradation and vitalization of polypropylene along with jute fibers present in composites and the residue that are formed after degradation requires higher temperature for subsequent degradation. The initial (at 10% and 20% weight loss), major (at 75% weight loss) and final weight loss and their corresponding degradation temperatures for the jute-PP, kenaf-PP, and PALF-PP composites and 100% PP are given in Table 4.

From Figure 11a, it is clear that major weight loss (75%) of the 35/65 Jute-PP composite occurred at 426 °C possibly due to degradation of the PP along with the jute fibers present in the composite. Whereas the final weight loss (~95–98% loss) of this composite sample (35/65 Jute/PP) occurred at 53 °C which is higher than that of the 100% PP. This final degradation of the jute-PP composite shifted to higher temperature due to a stronger adhesion between jute fibers and the PP matrix. Hence, the effect of this jute fiber-reinforced composite (35/65) showed increased thermal stability at higher temperature compared to other jute fiber wt.%. Similar type of thermal stability was also reported in case of jute-reinforced unidirectional epoxy composite [52] as well as short jute fiber-reinforced PP composite [53].

The TGA results of kenaf fiber-reinforced polypropylene composite at 30, 35, and 40 wt.% fiber loading are illustrated in Figure 11b. The major weight loss (75%) of the 40/60 Kenaf/PP composite occurred at 424 °C due to degradation of the PP matrix along with the kenaf fibers. The degradation of final residue i.e., final weight loss (~95–98% loss) of this composite sample (40/60 kenaf/PP) occurred at 544 °C which was higher than that of the 100% PP. It should be noted that the degradation temperatures (at different weight loss%) do not change significantly with varying kenaf fiber loading in the composites. Azam et el. [27] reported that with the increase in kenaf fibers in the PP matrix materials, the activation energy increases.

The TGA results of the PALF-PP composite at 30, 35, and 40 wt.% fiber loadings are shown in Figure 11c. The major weight loss (75%) of the 35/65 and 40/60 PALF-PP composites occurred at around 420 °C due to degradation of the PP as well as the PALF fibers. Here the final weight loss (~95–98% loss) of these composite samples (35/65 and 40/60 PALF/PP) occurred at 538 °C and 539 °C respectively which was higher than that of the 100% PP. Here the same reason can be given for the degradation of the PALF composite that shifted to a higher temperature like the other two composites.

Thus, in general, it can be revealed that all three fiber-reinforced PP composites possessed higher thermal stability than the pure PP particularly at major and final weight loss conditions. At final degradation condition, both kenaf-PP and PALF-PP composites showed an increase in degradation temperature with the fiber content. The thermal stability performance of the composites can be ranked with the following ascending order jute-PP < kenaf-PP < PALF-PP. The difference in constituent (like cellulose and lignin) of the fibers may affect the thermal stability of the composites [36].

### 3.7. Fractured Surface Morphology

SEM images of the fractured composite surfaces during tensile testing showed general characteristics of fiber breaking and fiber pull out (Figure 12). Furthermore, the broken fibers were clean indicating a lack of adhesion. Similar types of fiber pull out was reported by Shahinur et al. [46] for jute-PP and treated jute-PP composite.

Closer look at the interface at high magnification also revealed gaps between the fibers and the matrix for all composites (Figure 13). However, the gap was relatively smaller in the case PALF-PP composites indicating a better fiber matrix adhesion, which could be responsible for bearing higher load. As the natural fibers are different in their structures, their lumen sizes and shapes are also different. Therefore, the product density will be different. From the SEM fractured surfaced, it was clear that the interaction between the fiber matrix and penetration of the matrix material varied in the composite depending on the fiber type, which affected the gap formation and mechanical properties of the composite.

Therefore, the distinctive surface characteristics of the PALF fiber leading to a better adhesion with the PP matrix and the relatively higher tensile strength of the PALF fiber could be attributed to the higher strength displayed by the PALF-PP composites compared to the other two composites.

## 4. Conclusions

Polypropylene (PP)-based composites reinforced with three natural fibers such as, jute, kenaf, and pineapple leaf fiber (PALF) were prepared, and their structural, mechanical, and thermal properties were evaluated. FTIR study indicated subtle differences in the fiber structures of the composites with no strong evidence of chemical interaction between the fibers and composites. Overall, all the composites showed better mechanical properties compared to the pure PP, but for all composites improvement in tensile strengths were higher than the flexural strengths. The best mechanical properties were found for the PALF-PP composite at the higher fiber content (40 wt.%). In terms of impact strength, the composites can be ranked as PALF-PP > jute-PP > kenaf-PP where an increase in fiber content increased the impact strength in all the composites. Among the composites, improved properties in PALF-PP composites could be related to its higher fiber strength and better interfacial bonding as evidenced by the SEM images of the fractured surfaces. Furthermore, a good agreement was obtained between the experimental and the theoretical results of the tensile modulus of short fiber composites obtained by Pan, 1994 model, whereas Miao and Shan, 2011 model clearly overestimated the experimental results. The composite samples displayed better thermal stability particularly at higher degradation temperature. From this study, the natural fibers can be selected to fabricate the final composite materials as per the end application requirements. The effect of aging (physical, thermal, and mechanical) on the mechanical properties of the fiber-reinforced composites will be explored in the near future.

## Figures and Tables

**Figure 1 polymers-15-00830-f001:**
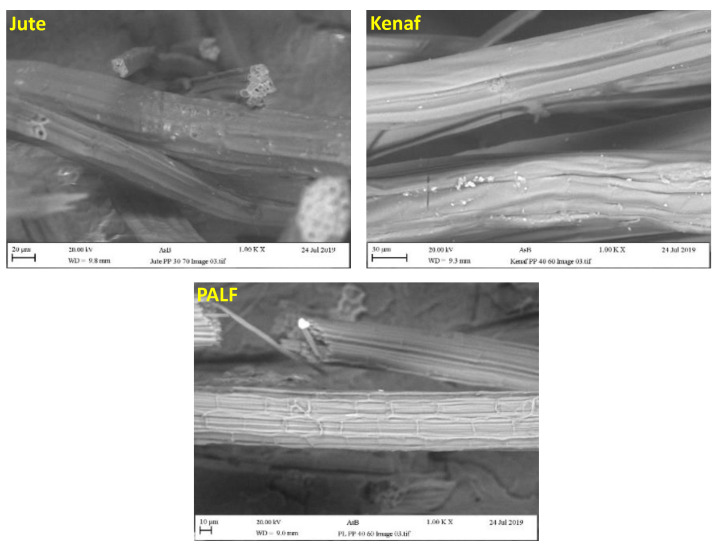
Surface morphologies of jute, kenaf, and PALF fibers.

**Figure 2 polymers-15-00830-f002:**
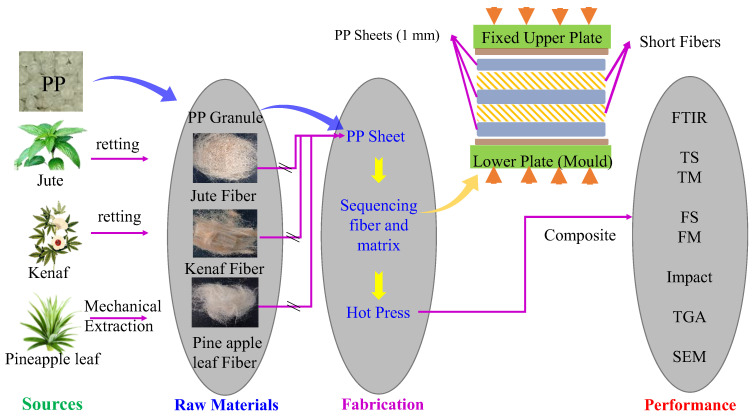
Schematic diagram of experimental procedure.

**Figure 3 polymers-15-00830-f003:**
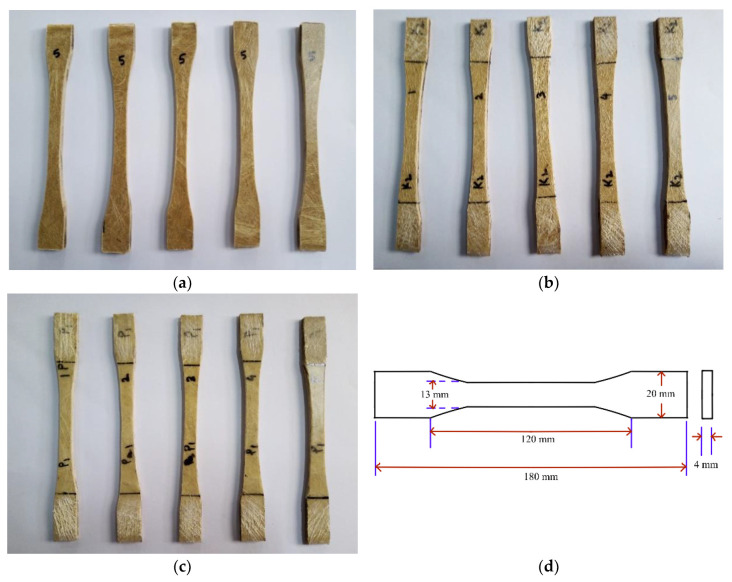
Composite specimens with (**a**) jute (**b**) kenaf, (**c**) PALF fibers before tensile testing and (**d**) specimen dimension: 180 mm × 20 mm × 4 mm.

**Figure 4 polymers-15-00830-f004:**
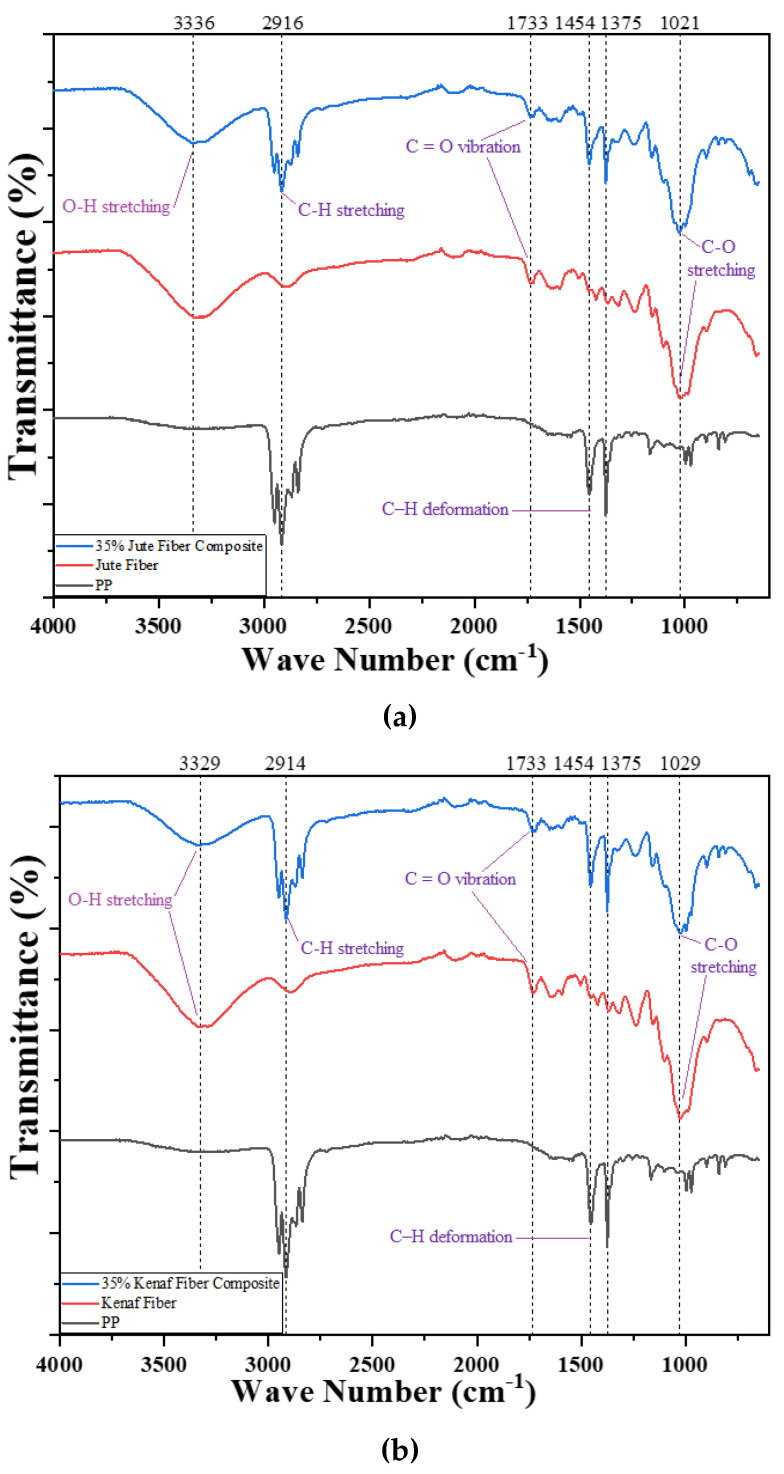
FTIR spectra of the PP, fibers and composites reinforced with 35 wt.% (**a**) jute, (**b**) kenaf, and (**c**) PALF.

**Figure 5 polymers-15-00830-f005:**
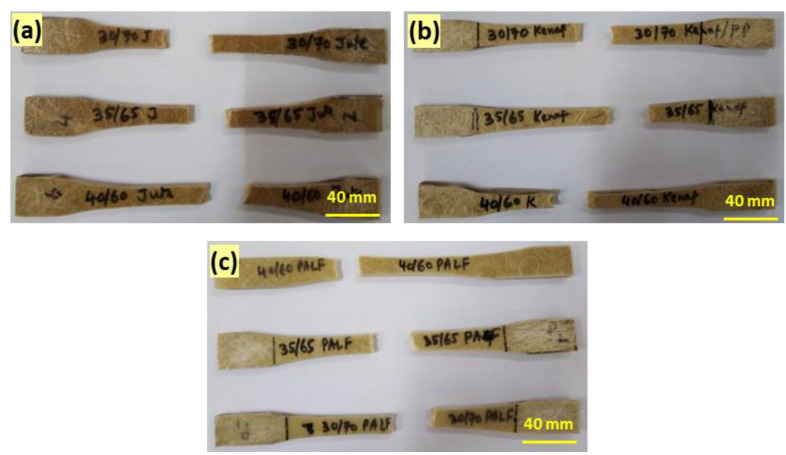
Tensile testing of composite specimens with (**a**) jute, (**b**) kenaf, and (**c**) PALF fibers.

**Figure 6 polymers-15-00830-f006:**
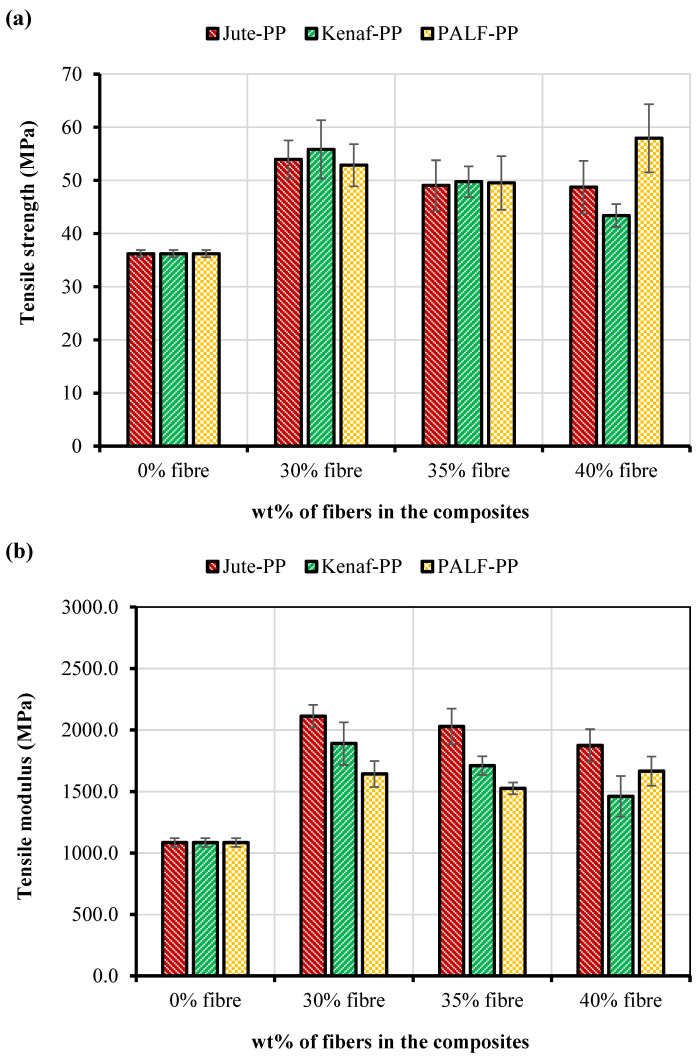
(**a**) Tensile strengths, (**b**) tensile moduli of the composite specimens.

**Figure 7 polymers-15-00830-f007:**
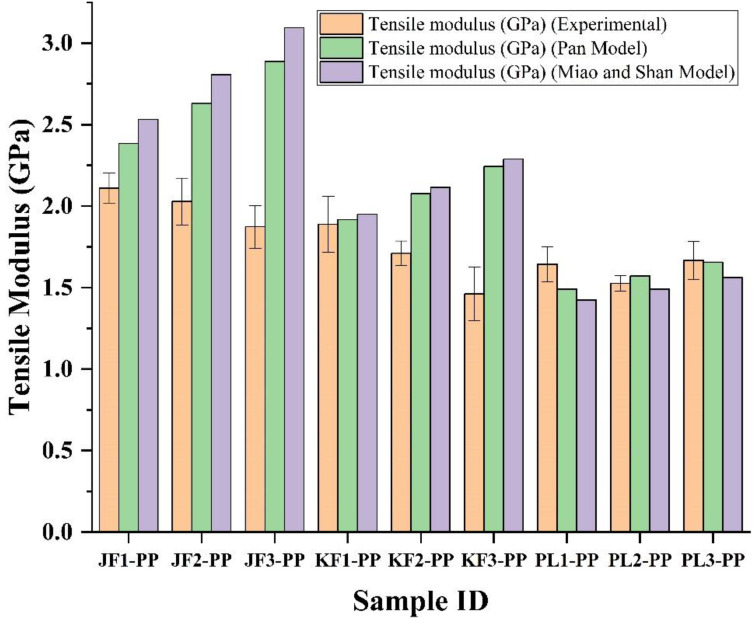
Comparison between theoretical and experimental results of elastic modulus of jute, kenaf, and PALF fibers-reinforced PP composites having different weight proportions of fibers.

**Figure 8 polymers-15-00830-f008:**
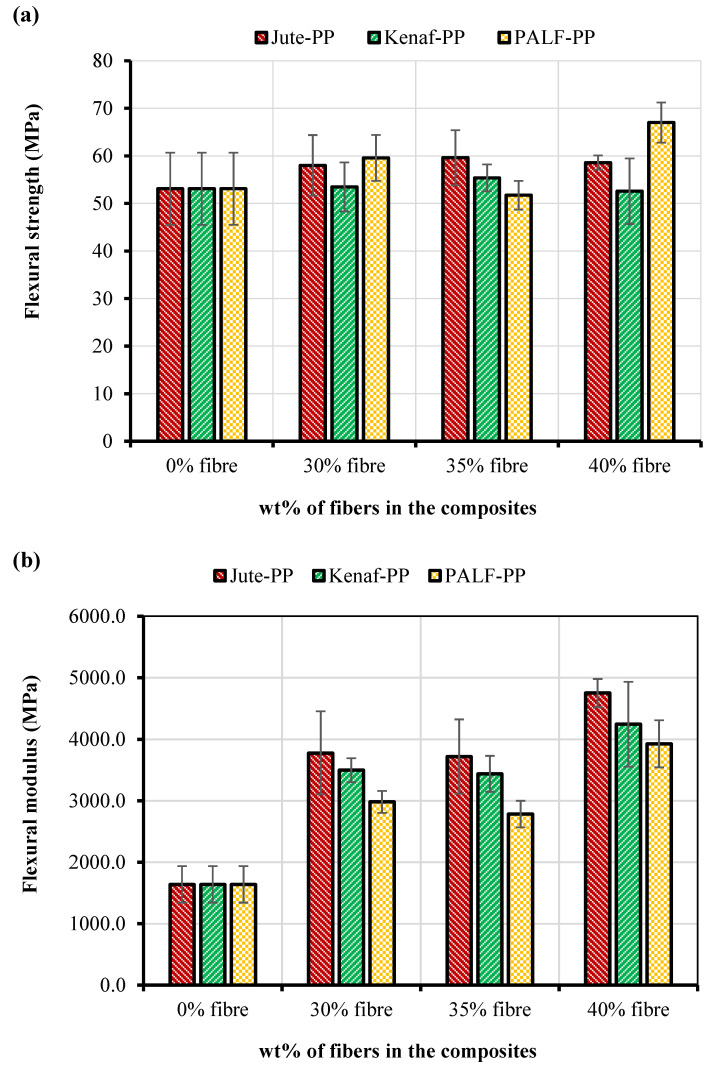
(**a**) Flexural strengths and (**b**) modulus of jute, kenaf, and PALF fiber-reinforced PP composite specimens.

**Figure 9 polymers-15-00830-f009:**
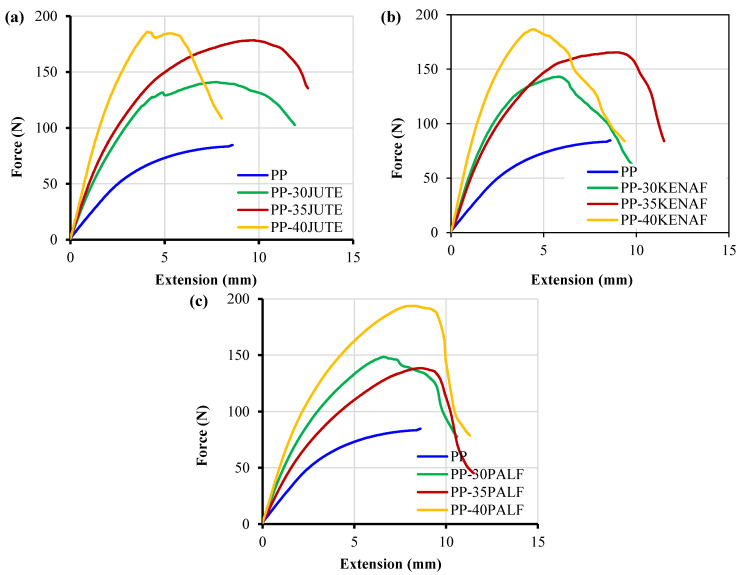
Force vs. extension graphs during flexural testing of the composite specimens (**a**) jute, (**b**) kenaf, and (**c**) PALF.

**Figure 10 polymers-15-00830-f010:**
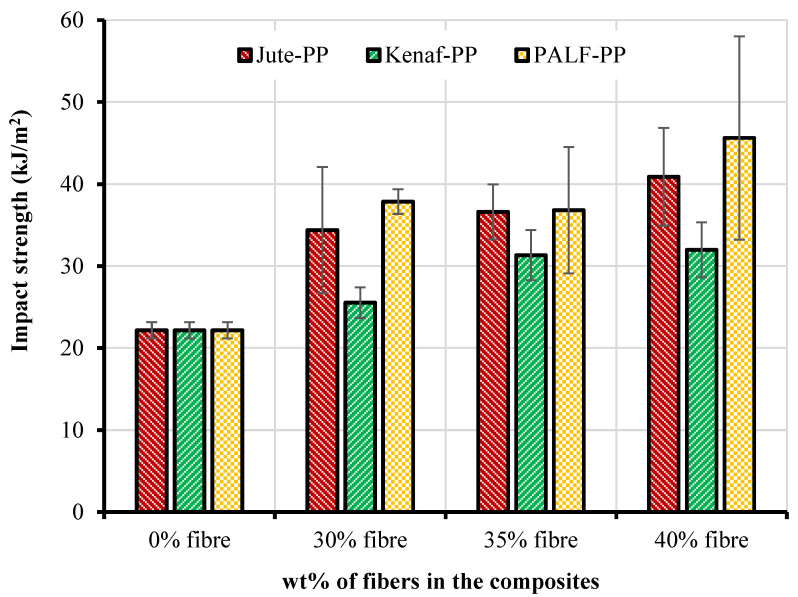
Impact strengths of the natural fiber-reinforced PP composites.

**Figure 11 polymers-15-00830-f011:**
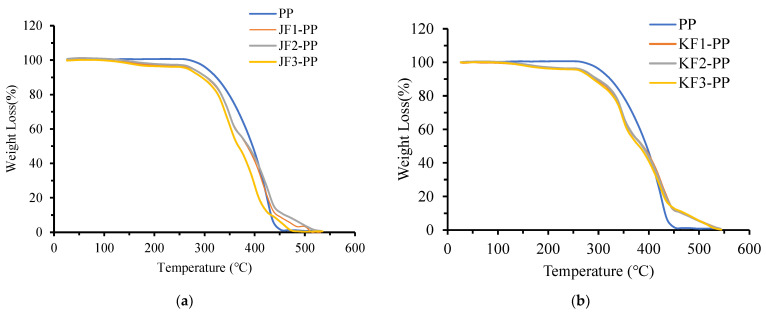
TGA curves of (**a**) jute, (**b**) kenaf, and (**c**) PLAF-PP composites at 30, 35, and 40 wt.% fiber loadings.

**Figure 12 polymers-15-00830-f012:**
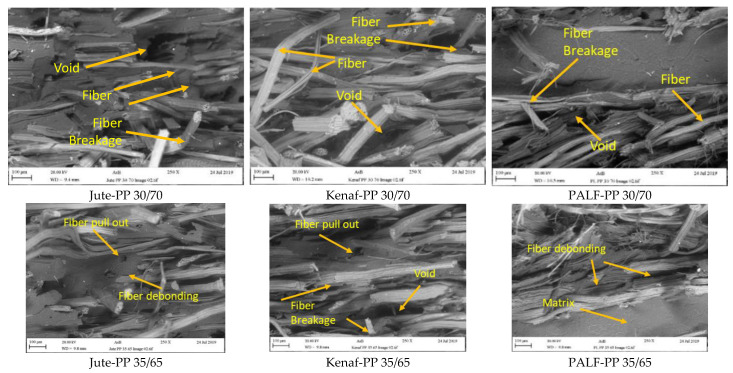
SEM images of fractured composites reinforced with different wt.% of jute, kenaf and PLAF fibers.

**Figure 13 polymers-15-00830-f013:**
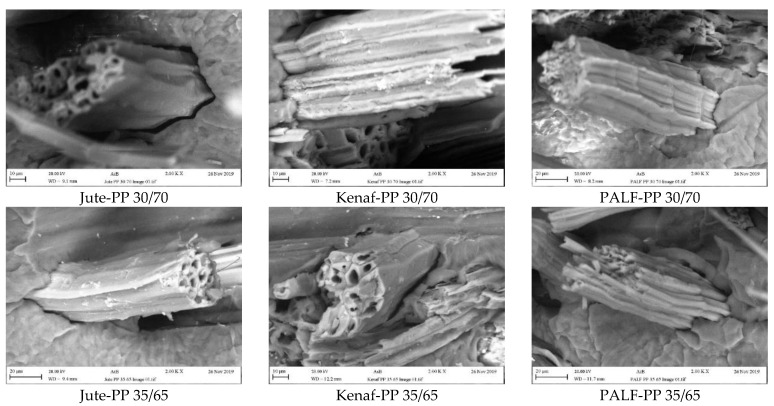
Close view of fiber matrix interaction in all the composites.

**Table 1 polymers-15-00830-t001:** Jute, kenaf, and pineapple leaf fiber characterization parameter with composite fabrication method.

Fibers	Composite	Mechanical/Range	Fiber Treatment	Fiber%	Fabrication	Ref.
Jute	Oxidized Jute-PP	TS increased (31–24 MPa), TM decreased (1.6–2.3 GPa), FS increased (44–56 GPa), IS increased (31–45 J/m), hardness increased (77–90)	Formic acid (HCOOH), sodium periodate solution, urea [CO(NH_2_)_2_]	20–35%	Single screw extrusion	[19]
	Jute fabrics-PP	TS (68.1 MPa) and BS (94.1 MPa) increased TM (2936 MPa) and BM (4831 MPa) increased, chemical absorption was high, IS increased	-	30–55%	Compression molding	[20]
	Jute short fiber-PP	Water absorption rate was high for alkali treated composites, TS increased (21–30 MPa), TM increased (1.3–3 GPa)	Alkali, potassium permanganate, and silane	30%	Twin Screw extrusion	[21]
	Jute fabrics-PP	TS increased (20.48–47.08 MPa), TM increased (1.24–3.58 GPa),	Nonpolar octyl gallate (OG), dodecyl gallate (DG), and octadecyl gallate (OCG)	50%	Hot press	[22]
	PP/jute	TS and FS decreased with the addition of ESO and TOA, IS increased with the addition of ESO and TOA,	Epoxy soybean oil (ESO)/tung oil anhydride (TOA)		Torque rheometer followed by injection molded	[23]
	Jute mat-PP-MAPP	TS increased for 30% MAPP (11.62–26.91 MPa) highest TM value shown for 20% MAPP (1590.73 MPa), FS increased (38.45–6.95 MPa)	-	20–30%	Compression molding	[24]
	Jute fiber-PP	Storage modulus increased (580–1600 MPa), 2 ply of 3 cm jute fiber showed the highest TS (17.86 MPa),	-	0–10%	Hot press	[25]
	Jute fiber-PP	Highest TS (25.8 MPa) and TM (1.7 GPa) for 40% jute-60%PP, highest FS (17.1 MPa) and FM (16.7 GPa) for 30% jute-70% PP, IS decreased (18–10KJ/m^2^),	-	30–60%	Compression molding	[26]
Kenaf (K)	MWCNT-K-PP	Viscosity decreased, TGA value is decreased (187–200 °C), water absorption stability increased, flammability is decreased, viscosity increased to 300–1600 Pa.S when MWCNT is added; when kenaf fiber incorporated viscosity increased 50–300 Pa	-	10–40%	Injection molding	[27]
	K-PP	The highest TS value (42 MPa) was shown for 5% NaOH treated 30% fiber content, TS value (58 MPa) increased for alkali–silane treated composites, TM (3 GPa) of alkali-silane treated composited was high than untreated and alkali treated composites Highest FS (55 MPa) was for 6% alkali-treated composites SEM examinations showed that TS and FS of composites increased for alkali treatment	NaOH, (alkali–silane treatment)		Compression molding	[28]
	K-PP	Highest TS (48 MPa) was shown for 30% kenaf–PP composites, and flax-PP composites showed the highest FS (76 MPa) and specific modulus was highest for kenaf–PP composite	-	30% and 40%	Compression molding	[29]
	K-PP-MH-MAgPP	Thermal stability decreased, TS decreased (40–23 MPa), TM (1.3–0.8) MPa increased, BS (65–108 MPa) decreased, and BM (6–10 GPa) increased with fiber content; with the addition of MgOH, TS (22–23 MPa) decreased, and BM (5–7 GPa) increased	-	10–25%	Haake RheocordRPM is 50	[30]
	K-PP	TS increased(25–50 MPa), TM increased (1–3 GPa), FS increased (31–70 MPa), FM increased (1.2–3.1 GPa), IS decreased (5.8–4.7 KJ/m^2^)	NaOH	10–40%	Close molding injection	[31]
	K-PP	Correlation measured between the physical and mechanical properties	Alkaline, silane	-	Heat extrusion	[32]
	K-CNT-PP	Higher TS (16 MPa) shown for 30 wt%, TM increased (700–2200 MPa), FS increased (22–26 MPa), and FM increased (600–2400 MPa)	-	20–40%	Injection molding	[33]
	K-PP	TS (158–85 MPa) decreased, TM (12200–7000 MPa) decreased with orientation of fiber and temperature (30–120 °C); BS (15–105 MPa) decreased, BM (1–8 GPa) decreased, and storage modulus (10–80 GPa) decreased and Poisson’s ratio (0.05–0.45)	-	40%	Compression molding	[34]
PALF	PALF- LDPE	TS (37–40 MPa), IS, BS, thermal stability are highest for 7% NaOH and 7.5 Gamma radiation, TM (1–1.6 GPa), TM (1.3–1.6 GPa), BS (91–97) MPa,	NaOH, gamma	50%	Heat press	[35]
	PALF-PP	Highest TS (42.2 MPa) and TM (1864 MPa) for untreated composites, TS (55.9 MPa) increased for ZnCl_2_ treatment, highest FS (51.6 MPa) shown for 40/60 weight percentage and for HNO_3_ treatment	Sodium Hydroxide (NaOH) solution, Zinc chloride, Acetic Anhydride and Nitric acid	10–40%	Injection molding	[36]
	PALF-PP	TS (14.98 MPa) for 30% PALF	-	10–30%	Twin-screw extrusion	[37]
	PALF-LDPE	TS increased (17–28 MPa), TM increased (400–800 MPa), BS increased (54–78 MPa), BM increased (1000–5800 MPa), highest impact strength was 33 KJ/m^2^ for 50% fiber weight	Gamma radiation	10–60%	Compression molding	[38]
	PALF-PP	TS increased (28–87 MPa) TM increased (338–1731 MPa), BS increased (20–51 MPa), BM increased (230–840 MPa), IS increased (2.9–7.2 KJ/m^2^),	NaOH	25–45%	Compression molding	[39]
	PALF-TBP	TS increased upto18.37 MPa with 30% fiber and TM increased to 1.03 GPa, BS increased to 19.34 MPa, IS increased to 18.10 kJ/m^2^ with 40% fiber	-	10–40%	Compression molding	[40]
	PALF-PP	With 20 wt% PALF fibers, increase Young’s modulus (146%) and stress at break (112%), but decrease in elongation at break (298%)	-	-	Twin-screw extrusion	[41]
	PALF-PP/LDPE	Highest TS (54 MPa) for 15/85% PALF-PP composites and increased with the increase of PALF fiber	NaOH	0–25%	Compression molding	[42]
	PALF-PP	TS (37.28 MPa) and TM (687.02 MPa) for 10.8% fiber content, FM (2000 MPa) was higher for 2.7% fiber content	-	0–18%	Hot press	[43]

Note: TBP—tapioca biopolymer, LDPE—low density polyethylene, MAgPP—maleic anhydride–grafted polypropylene, MWCNT—multi-wall carbon nanotube, TS—tensile strength, F/BS—flexural/bending strength, TM—tensile modulus, BM—bending modulus, IS—impact strength, TGA—thermal gravimetric analysis, DMA—dynamic mechanical analysis, DSC—differential scanning calorimetry, SEM—scanning electron microscope.

**Table 2 polymers-15-00830-t002:** Physical and mechanical properties of jute fiber, kenaf fiber, PALF, and PP.

Types of Fiber/Matrix	Diameter (µm)	Density (gm/cm^3^)	Tensile Strength (MPa)	Tensile Modulus (GPa)
Jute	53.38 ± 5.93	1.40	300–773	20–55
Kenaf	55–60	1.45	350–600	26.00
PALF	20–40	1.56	413–1627	60–82
PP	-	0.91	36.21 ± 0.68	1.085 ± 0.036

**Table 3 polymers-15-00830-t003:** Fiber weight, volume fractions, and layer configuration of the fiber-reinforced PP composites.

Samples ID	Code	Fiber wt.%	Fiber V*_f_*%
30/70 Jute/PP Composite	JF1-PP	30	21.60
35/65 Jute/PP Composite	JF2-PP	35	25.71
40/60 Jute/PP Composite	JF3-PP	40	30.00
30/70 Kenaf/PP Composite	KF1-PP	30	20.45
35/65 Kenaf/PP Composite	KF2-PP	35	24.41
40/60 Kenaf/PP Composite	KF3-PP	40	28.57
30/70 PALF/PP Composite	PL1-PP	30	26.49
35/65 PALF/PP Composite	PL2-PP	35	31.17
40/60 PALF/PP Composite	PL3-PP	40	35.92

**Table 4 polymers-15-00830-t004:** TGA data of jute, kenaf and PALF-reinforced PP composites.

ID	Degradation Temperature, °C (T_10%_)	Degradation Temperature, °C (T_20%_)	Degradation Temperature,°C (T_75%_)	Degradation Temperature, °C (at T_98%_)
100% PP	323.31	348.51	420.23	500.11
JF1-PP	302.26	331.94	420.04	500.21
JF2-PP	303.60	332.80	426.01	531.57
JF3-PP	293.97	325.98	402.45	488.41
KF1-PP	294.86	331.17	428.24	541.17
KF2-PP	297.38	332.20	427.05	542.79
KF3-PP	289.67	327.46	424.09	544.13
PL1-PP	298.70	325.51	409.64	504.86
PL2-PP	295.01	327.46	422.61	538.35
PL3-PP	292.93	327.02	424.24	539.53

Note: T_10%_ = degradation temperature at 10% weight loss; T_20%_ = degradation temperature at 20% weight loss; T_75%_ = degradation temperature at 75% weight loss, T_98%_ = degradation temperature at 98% weight loss.

## Data Availability

The data presented in this study are available within the article.

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
