# Peer review of "Assessing Mechanical Properties of Jute, Kenaf, and Pineapple Leaf Fiber-Reinforced Polypropylene Composites: Experiment and Modelling"

_polymers, 2023, doi:10.3390/polym15040830_

Round 1

Reviewer 1 Report

The problem of filling polymers with materials of natural origin is well known and arouses considerable interest among researchers. There is still a lot to explore and understand about this topic. The authors fit well into current research trends. However, it is worth asking the question here, what about such products, what is the issue related to their recycling? What are the chances of reducing the release of microplastics?

1.       The subject of the work is consistent with its content, the introduction gives a good background for the set goal.

2.       The literature review is extensive, up-to-date and well-chosen.

3.       Table 1 is a bit confusing, it would be worth simplifying it somehow, but I understand that the authors wanted to present all the data as best as possible.

4.       The adopted methodology is good, the authors demonstrated good research skills. The manuscript itself is extensive and could be divided into two.

5.       Figures and photo are well described and complete the body of the manuscript.

6.       The final conclusions are well formulated, they could be slightly extended, but they result from the conducted research and justify the purpose of the work.

Reviewer 2 Report

We put the reviewer comment in the attachment.

Reviewer 3 Report

The article "Assessing Mechanical Properties of Jute, Kenaf and Pineapple Leaf Fiber Reinforced Polypropylene Composites: Experiment and Modeling" is a professionally structured information on a hot topic related to the creation of composites with additives of natural fibers. The authors very briefly stated the problem of this scientific direction and described the results of the study of three types of natural fibers: jute, kenaf and pineapple leaf fiber (PALF). At the same time, the main characteristics of the fibers, necessary from their point of view, were given (Table 2). The results obtained are illustrated and truthfully described. The strength of the article is the brevity of the presentation of the bulk experimental material for polypropylene composites with the addition of three different types of natural fibers. The article cites 54 primary sources, including books, monographs and articles published in publications devoted to both natural fibers and composites. By title, abstract, research methods, main results, the article refers specifically to Polymers. The authors presented the prediction of the properties of composites based on the ratio of polypropylene and natural fibers. These results may be the subject of discussion and will attract potential readers. An analysis of the properties of natural fibers suggests the distinctive properties of the composite with the addition of PALF. The article is recommended for publication even in its present form, but let me point out the shortcomings.

Drawbacks to be addressed:

1. Align sections 2 and 3 with each other so that the discussion of the results does not begin with a description of the IR spectra (2.6. Fourier Transform Infrared (FTIR) Spectroscopy).

2. Please add to the text the author's version (explanation) of the distinctive properties of PALF in composites.

3. The format of the article allows citing reviews and experimental articles on this topic published in MDPI.
